# LightSpeed: Light and Fast
# Neural Light Fields on Mobile Devices

**Aarush Gupta**[1]    **Junli Cao**[2,†]    **Chaoyang Wang**[2,†]    **Ju Hu**[2]    **Sergey Tulyakov**[2]
**Jian Ren**[2]    **László A Jeni**[1]
[1]Robotics Institute, Carnegie Mellon University    [2]Snap Inc.
Project page: https://lightspeed-r2l.github.io

## Abstract

Real-time novel-view image synthesis on mobile devices is prohibitive due to the limited computational power and storage. Using volumetric rendering methods, such as NeRF and its derivatives, on mobile devices is not suitable due to the high computational cost of volumetric rendering. On the other hand, recent advances in neural light field representations have shown promising real-time view synthesis results on mobile devices. Neural light field methods learn a direct mapping from a ray representation to the pixel color. The current choice of ray representation is either stratified ray sampling or Plücker coordinates, overlooking the classic light slab (two-plane) representation, the preferred representation to interpolate between light field views. In this work, we find that using the light slab representation is an efficient representation for learning a neural light field. More importantly, it is a lower-dimensional ray representation enabling us to learn the 4D ray space using feature grids which are significantly faster to train and render. Although mostly designed for frontal views, we show that the light-slab representation can be further extended to non-frontal scenes using a divide-and-conquer strategy. Our method offers superior rendering quality compared to previous light field methods and achieves a significantly improved trade-off between rendering quality and speed.

## 1 Introduction

Real-time rendering of photo-realistic 3D content on mobile devices such as phones is crucial for mixed-reality applications. However, this presents a challenge due to the limited computational power and memory of mobile devices. The current graphics pipeline requires storing tens of thousands of meshes for complex scenes and performing ray tracing for realistic lighting effects, which demands powerful graphics processing power that is not feasible on current mobile devices. Recently, neural radiance field (NeRF [23]) has been the next popular choice for photo-realistic view synthesis, which offers a simplified rendering pipeline. However, the computational cost of integrating the radiance field remains a bottleneck for real-time implementation on mobile devices. There have been several attempts to reduce the computational cost of this integration step, such as using more efficient radiance representations [13, 40, 28, 17, 5, 10] or distilling meshes from radiance field [34, 6, 39, 35, 27, 29]. Among these approaches, only a handful of mesh-based methods [6, 29] have demonstrated real-time rendering capabilities on mobile phones, but with a significant sacrifice in rendering fidelity. Moreover, all aforementioned methods require significant storage space (over 200MB), which is undesirable for mobile devices with limited onboard storage.

---

†These authors contributed equally.

37th Conference on Neural Information Processing Systems (NeurIPS 2023).

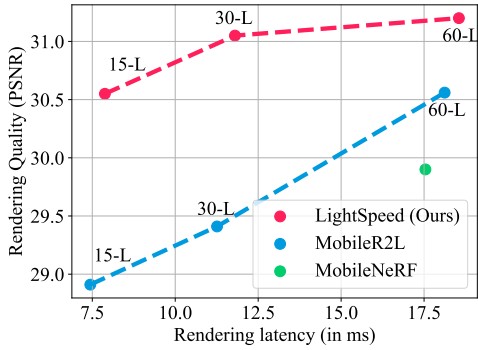 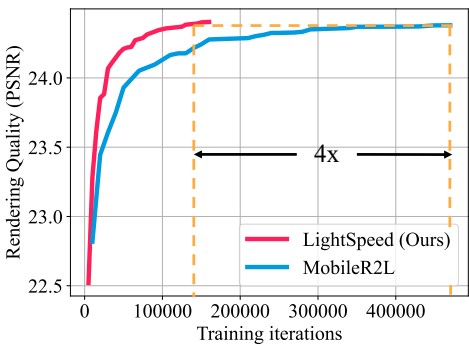

(a) Rendering latency v/s fidelity.

(b) Faster training speed.

Figure 1: Our LightSpeed approach demonstrates a superior trade-off between on-device rendering quality and latency while maintaining a significantly reduced training time and boosted rendering quality. **(a)** rendering quality and latency on the $400 \times 400$ Lego scene [23] running on an iPhone 13. **(b)** training curves for the $756 \times 1008$ Fern scene [22].

Alternatively, researchers have used 4D light field[1] (or lumigraph) to represent radiance along rays in empty space [11, 24, 12, 19], rather than attempting to model the 5D plenoptic function as in NeRF-based approaches. Essentially, the light field provides a direct mapping from rays to pixel values since the radiance is constant along rays in empty space. This makes the light field suitable for view synthesis, as long as the cameras are placed outside the convex hull of the object of interest. Compared to integrating radiance fields, rendering with light fields is more computationally efficient. However, designing a representation of light field that compresses its storage while maintaining high view-interpolation fidelity remains challenging. Previous methods, such as image quilts [38] or multiplane images (MPI) [41, 16, 32, 9], suffer from poor trade-offs between fidelity and storage due to the high number of views or image planes required for reconstructing the complex light field signal. Recent works [36, 4, 2, 31] have proposed training neural networks to represent light fields, achieving realistic rendering with a relatively small memory footprint. Among those, MobileR2L [4] uses less than 10MB of storage per scene, and it is currently the only method that demonstrates real-time performance on mobile phones.

However, prior neural light field (NeLF) representations, including MobileR2L, suffer from inefficiencies in learning due to the high number of layers (over 60 layers), and consequently, a long training time is required to capture fine scene details. One promising strategy to address this issue is utilizing grid-based representations, which have proven to be effective in the context of training NeRFs [30, 25, 17, 10]. Nonetheless, incorporating such grid-based representation directly to prior NeLFs is problematic due to the chosen ray parameterization. R2L [36] and MobileR2L [4] parameterize light rays using a large number of stratified 3D points along the rays, which were initially motivated by the discrete formulation of integrating radiance. However, this motivation is unnecessary and undermines the simplicity of 4D light fields because stratified sampling is redundant for rays with constant radiance. This becomes problematic when attempting to incorporate grid-based representations for more efficient learning, as the high-dimensional stratified-point representation is not feasible for grid-based discretization. Similarly, the 6-dimensional Plücker coordinate used by Sitzmann *et al.* [31] also presents issues for discretization due to the fact that Plücker coordinates exist in a projective 5-space, rather than Euclidean space.

In this paper, we present *LightSpeed*, the first NeLF method designed for mobile devices that uses a grid-based representation. As shown in Fig. 1, our method achieves a significantly better trade-off between rendering quality and speed compared to prior NeLF methods, while also being faster to train. These advantages make it well-suited for real-time applications on mobile devices. To achieve these results, we propose the following design choices:

**First**, we revisit the classic 4D light-slab (or two-plane) representation [12, 19] that has been largely overlooked by previous NeLF methods. This lower-dimensional parameterization allows us to compactly represent the rays and efficiently represent the light field using grids. To our knowledge,

---

[1]For the rest of the paper, we will use the term 'light field' to refer to the 4D light field, without explicitly stating the dimensionality.

Attal *et al*. [2] is the only other NeLF method that has experimented with the light-slab representation. However, they did not take advantage of the grid-based representation, and their method is not designed for real-time rendering. **Second**, to address the heavy storage consumption of 4D light field grids, we take inspiration from k-planes [10] and propose decomposing the 4D grids into six 2D feature grids. This ensures that our method remains competitive for storage consumption compared to prior NeLF methods. **Third**, we apply the super-resolution network proposed by MobileR2L [4], which significantly reduces the computational cost when rendering high-resolution images. **Finally**, the light-slab representation was originally designed for frontal-view scenes, but we demonstrate that it can be extended to represent non-frontal scenes using a divide-and-conquer strategy.

Our contributions pave the way for efficient and scalable light field representation and synthesis, making it feasible to generate high-quality images of real-world objects and scenes. Our method achieves the highest PSNR and among the highest frame rates (55 FPS on iPhone 14) on LLFF (frontal-view), Blender ($360°$), and unbounded $360°$ scenes, proving the effectiveness of our approach.

## 2 Related work

**Light Field.** Light field representations have been studied extensively in the computer graphics and computer vision communities [38]. Traditionally, light fields have been represented using the 4D light slab representation, which parameterizes the light field by two planes in 4D space [12, 19]. More recently, neural-based approaches have been developed to synthesize novel views from the light field, leading to new light field representations being proposed.

One popular representation is the multi-plane image (MPI) representation, which discretizes the light field into a set of 2D planes. The MPI representation has been used in several recent works, including [41, 16, 32, 9, 7]. However, the MPI representation can require a large amount of memory, especially for high-resolution light fields. Another recent approach that has gained substantial attention is NeRF [23] (Neural Radiance Fields), which can synthesize novel views with high accuracy, but is computationally expensive to render and train due to the need to integrate radiance along viewing rays. There has been a substantial amount of works [37, 26, 28, 21, 13, 40, 28, 17, 5, 10, 34, 6, 39, 35, 27, 29, 36, 4, 2, 31] studying how to accelerate training and rendering of NeRF, but in the following, we focus on recent methods that achieve real-time rendering with or without mobile devices.

**Grid Representation of Radiance Field.** The first group of methods trade speed with space, by precomputing and caching radiance values using grid or voxel-like data structures such as sparse voxels [30, 13], octrees [40], and hash tables [25]. Despite the efficient data structures, the memory consumption for these methods is still high, and several approaches have been proposed to address this issue. First, Chen *et al*. [5] and Fridovich-Keil *et al*. [10] decompose voxels into matrices that are cheaper to store. Takikawa *et al*. [33] performs quantization to compress feature grids. These approaches have enabled real-time applications on desktop or server-class GPUs, but they still require significant computational resources and are not suitable for resource-constrained devices such as mobile or edge devices.

**Baking High Resolution Mesh.** Another group of methods adopts the approach of extracting high-resolution meshes from the learned radiance field [6, 29, 35]. The texture of the mesh stores the plenoptic function to account for view-dependent rendering. While these approaches have been demonstrated to run in real-time on mobile devices, they sacrifice rendering quality, especially for semi-transparent objects, due to the mesh-based representation. Additionally, storing high-resolution meshes with features is memory-intensive, which limits the resolution and complexity of the mesh that can be used for rendering.

**Neural Light Fields.** Recent works such as R2L [36], LFNS [31] and NeuLF [20] have framed the view-synthesis problem as directly predicting pixel colors from camera rays, making these approaches fast at inference time without the need for multiple network passes to generate a pixel color. However, due to the complexity of the 4D light field signal, the light field network requires sufficient expressibility to be able to memorize the signal. As a result, Wang *et al*. [36] end up using as many as 88 network layers, which takes three seconds to render one $200 \times 200$ image on iPhone 13. In this regard, Cao *et al*. [4] introduce a novel network architecture that dramatically reduces R2L's computation through super-resolution. The deep networks are only evaluated on a low-resolution ray bundle and then upsampled to the full image resolution. This approach, termed MobileR2L, achieves real-time rendering on mobile phones. NeuLF [20] also proposes to directly regress pixel colors

using a light slab ray representation but is unable to capture fine-level details due to lack of any sort of high-dimensional input encoding and is limited to frontal scenes. Another notable work, SIGNET [8], utilizes neural methods to compress a light field by using a ultra spherical input encoding to the light slab representation. However, SIGNET doesn't guarantee photorealistic reconstruction and hence deviates from task at hand. Throughout the paper, we will mainly compare our method to MobileR2L [4], which is currently the state-of-the-art method for real-time rendering on mobile devices and achieves the highest PSNR among existing methods.

It is important to note that training NeLFs requires densely sampled camera poses in the training images and may not generalize well if the training images are sparse, as NeLFs do not explicitly model geometry. While there have been works, such as those by Attal *et al.* [2], that propose a mixture of NeRF and local NeLFs, allowing learning from sparse inputs, we do not consider this to be a drawback since NeLFs focus on photo-realistic rendering rather than reconstructing the light field from sparse inputs, and they can leverage state-of-the-art reconstruction methods like NeRF to create dense training images. However, it is a drawback for prior NeLFs [36, 4] that they train extremely slowly, often taking more than two days to converge for a single scene. This is where our new method comes into play, as it offers improvements in terms of training efficiency and convergence speed.

## 3 Methodology

### 3.1 Prerequisites

**4D Light Fields** or Lumigraphs are a representation of light fields that capture the radiance information along rays in empty space. They can be seen as a reduction of the higher-dimensional plenoptic functions. While plenoptic functions describe the amount of light (radiance) flowing in every direction through every point in space, which typically has five degrees of freedom, 4D light fields assume that the radiance is constant along the rays. Therefore, a 4D light field is a vector function that takes a ray as input (with four degrees of freedom) and outputs the corresponding radiance value. Specifically, assuming that the radiance $\mathbf{c}$ is represented in the RGB space, a 4D light field is mathematical defined as a function, *i.e.*:

$$\mathcal{F} : \mathbf{r} \in \mathbb{R}^M \mapsto \mathbf{c} \in \mathbb{R}^3, \tag{1}$$

where $\mathbf{r}$ is $M$-dimensional coordinates of the ray depending how it is parameterized.

Generating images from the 4D light field is a straightforward process. For each pixel on the image plane, we calculate the corresponding viewing ray $\mathbf{r}$ that passes through the pixel, and the pixel value is obtained by evaluating the light field function $\mathcal{F}(\mathbf{r})$. In this paper, our goal is to identify a suitable representation for $\mathcal{F}(\mathbf{r})$ that minimizes the number of parameters required for learning and facilitates faster evaluation and training.

**MobileR2L.** We adopt the problem setup introduced by MobileR2L [6] and its predecessor R2L [36], where the light field $\mathcal{F}(\mathbf{r})$ is modeled using neural networks. The training of the light field network is framed as distillation, leveraging a large dataset that includes both real images and images generated by a pre-trained NeRF. Both R2L and MobileR2L represent $\mathbf{r}$ using stratified points, which involves concatenating the 3D positions of points along the ray through stratified sampling. In addition, the 3D positions are encoded using sinusoidal positional encoding [23]. Due to the complexity of the light field, the network requires a high level of expressiveness to capture fine details in the target scene. This leads to the use of very deep networks, with over 88 layers in the case of R2L. While this allows for detailed rendering, it negatively impacts the rendering speed since the network needs to be evaluated for every pixel in the image.

To address this issue, MobileR2L proposes an alternative approach. Instead of directly using deep networks to generate high-resolution pixels, they employ deep networks to generate a low-resolution feature map, which is subsequently up-sampled to obtain high-resolution images using shallow super-resolution modules. This approach greatly reduces the computational requirements and enables real-time rendering on mobile devices. In our work, we adopt a similar architecture, with a specific focus on improving the efficiency of generating the low-resolution feature map.

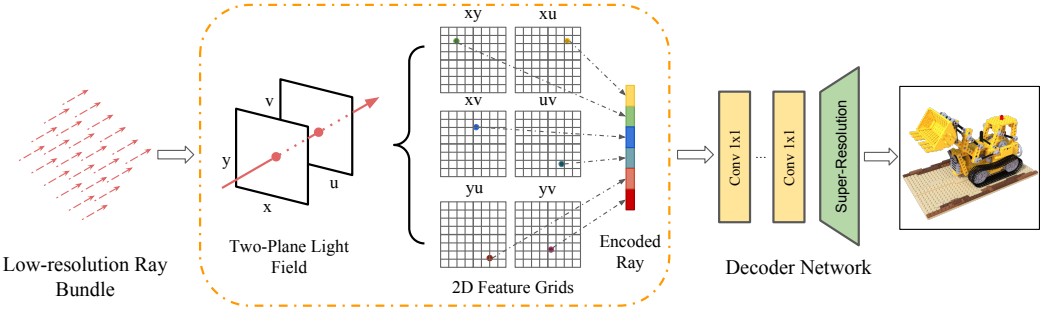

Figure 2: **LightSpeed Model for Frontal Scenes.** Taking a low-resolution ray bundle as input, our approach formulates rays in two-plane ray representation. This enables us to encode each ray using multi-scale feature grids, as shown. The encoded ray bundle is fed into a decoder network consisting of convolutions and super-resolution modules yielding the high-resolution image.

## 3.2 LightSpeed

We first describe the light-slab ray representation for both frontal and non-frontal scenes in Sec. 3.2.1. Next, we detail our grid representation for the light-slab in Sec. 3.2.2 and explain the procedure for synthesizing images from this grid representation in Sec. 3.3. Refer to Fig. 2 for a visual overview.

### 3.2.1 Ray Parameterization

**Light Slab (two-plane representation).** Instead of utilizing stratified points or Plücker coordinates, we represent each directed light ray using the classic two-plane parameterization[19] as an ordered pair of intersection points with two fixed planes. Formally,

$$\mathbf{r} = (x, y, u, v), \tag{2}$$

where $(x, y) \in \mathbb{R}^2$ and $(u, v) \in \mathbb{R}^2$ are ray intersection points with fixed planes $P_1$ and $P_2$ in their respective coordinate systems. We refer to these four numbers as the ray coordinates in the 4D ray space. To accommodate unbounded scenes, we utilize normalized device coordinates (NDC) and select the planes $P_1$ and $P_2$ as the near and far planes (at infinity) defined in NDC.

**Divided Light Slabs for Non-frontal Scenes.**   A single light slab is only suitable for modeling a frontal scene and cannot capture light rays that are parallel to the planes. To model non-frontal scenes, we employ a divide-and-conquer strategy by using a composition of multiple light slab representations to learn the full light field. We partition the light fields into subsets, and each subset is learned using a separate NeLF model. The partitions ensure sufficient overlap between sub-scenes, resulting in a continuous light field representation without additional losses while maintaining the frontal scene assumption. To perform view synthesis, we identify the scene subset of the viewing ray and query the corresponding NeLF to generate pixel values. Unlike Attal *et al.* [2], we do not perform alpha blending of multiple local light fields because our division is based on ray space rather than partitioning 3D space.

For *object-centric* 360° scenes, we propose to partition the scene into 5 parts using surfaces of a near-isometric trapezoidal prism and approximate each sub-scene as frontal (as illustrated in Fig. 3). For *unbounded* 360° scenes, we perform partitioning using k-means clustering based on camera orientation and position. We refer the reader to the supplementary material for more details on our choice of space partitioning.

### 3.2.2 Feature Grids for Light Field Representation

Storing the 4D light-slab directly using a high-resolution grid is impractical in terms of storage and inefficient for learning due to the excessive number of parameters to optimize. The primary concern arises from the fact that the 4D grid size increases quartically with respect to resolutions. To address this, we suggest the following design choices to achieve a compact representation of the light-slab without exponentially increasing the parameter count.

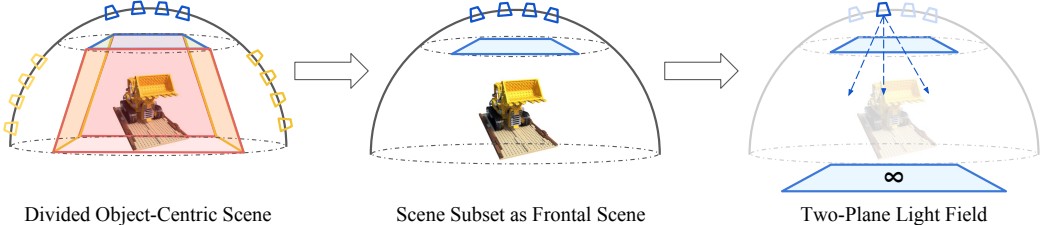

| Divided Object-Centric Scene | Scene Subset as Frontal Scene | Two-Plane Light Field |

Figure 3: **Space Partitioning for Non-frontal scenes.** We partition *object-centric* $360°$ scenes into 5 parts as shown. Each colored face of the trapezoidal prism corresponds to a partitioning plane. Each scene subset is subsequently learned as a separate NeLF

**Lower Resolution Feature Grids.** Instead of storing grids at full resolution, we choose to utilize low-resolution feature grids to take advantage of the quartic reduction in storage achieved through resolution reduction. We anticipate that the decrease in resolution can be compensated by employing high-dimensional features. In our implementation, we have determined that feature grids of size $128^4$ are suitable for synthesizing full HD images. Additionally, we adopt the approach from Instant-NGP [25] to incorporate multi-resolution grids, which enables an efficient representation of both global and local scene structures.

**Decompose 4D Grids into 2D Grids.** Taking inspiration from k-planes [10], we propose to decompose the 4D feature grid using $\binom{4}{2} = 6$ number of 2D grids, with each 2D grid representing a sub-space of the 4D ray space. This results in a storage complexity of $\mathcal{O}(6N^2)$, greatly reducing the storage required to deploy our grid-based approach to mobile devices.

### 3.3 View Synthesis using Feature Grids

Similar to MobileR2L [4], LightSpeed takes two steps to render a high resolution image (see Fig. 2).

**Encoding Low-Resolution Ray Bundles.** The first step is to render a low-resolution ($H_L \times W_L$) feature map from the feature grids. This is accomplished by generating ray bundles at a reduced resolution, where each ray corresponds to a pixel in a downsampled image. We project each ray's 4D coordinates $\mathbf{r} = (x, y, u, v)$ onto 6 2D feature grids $\mathbf{G}_{xy}, \mathbf{G}_{xu}, \mathbf{G}_{xv}, \mathbf{G}_{yu}, \mathbf{G}_{yv}, \mathbf{G}_{uv}$ to obtain feature vectors from corresponding sub-spaces. The feature values undergo bilinear interpolation from the 2D grids, resulting in six interpolated $F$-dimensional features. These features are subsequently concatenated to form a $6F$-dimensional feature vector. As the feature grids are multi-resolutional with $L$ levels, features $g_l(\mathbf{r}) \in \mathbb{R}^{6F}$ from different levels (indexed by $l$) are concatenated together to create a single feature $g(\mathbf{r}) \in \mathbb{R}^{6LF}$. Combining the features from all rays generates a low-resolution 2D feature map $\tilde{\mathbf{G}} \in \mathbb{R}^{H_L \times W_L \times 6LF}$, which is then processed further in the subsequent step.

**Decoding High-Resolution Image.** To mitigate the approximation introduced by decomposing 4D grids into 2D grids, the features $g(\mathbf{r})$ undergo additional processing through a MLP. This is implemented by applying a series of $1 \times 1$ convolutional layers to the low-resolution feature map . Subsequently, the processed feature map is passed through a sequence of upsampling layers (similar to MobileR2L [4]) to generate a high-resolution image.

## 4 Experiments

**Datasets.** We benchmark our approach on the real-world forward-facing [22] [23], the realistic synthetic $360°$ datasets [23] and unbounded $360°$ scenes [3]. The forward-facing dataset consists of 8 real-world scenes captured using cellphones, with 20-60 images per scene and 1/8th of the images used for testing. The synthetic $360°$ dataset has 8 scenes, each having 100 training views and 200 testing views. The unbounded $360°$ dataset consists of 5 outdoor and 4 indoor scenes with a central object and a detailed background. Each scene has between 100 to 300 images, with 1 in 8 images used for testing. We use $756 \times 1008$ LLFF dataset images, $800 \times 800$ resolution for the $360°$ scenes, and 1/4th of the original resolution for the unbounded $360°$ scenes.

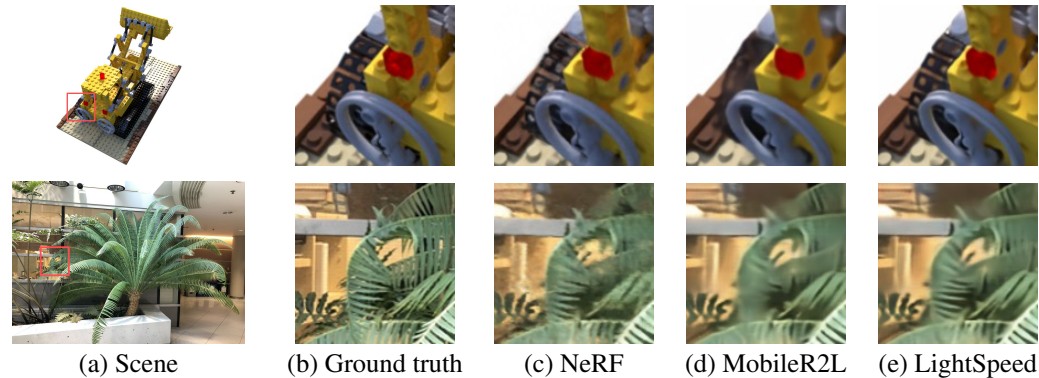

|          (a) Scene          |       (b) Ground truth       |       (c) NeRF       |       (d) MobileR2L       |       (e) LightSpeed       |

Figure 4: **Qualitative Results** on frontal and non-frontal scenes. Zoomed-in comparison between NeRF [23], MobileR2L [4] and our LightSpeed approach.

**Training Details.** We follow a similar training scheme as MobileR2L: train the LightSpeed model using pseudo-data mined from a pre-trained NeRF teacher. We specifically train MipNeRF teachers to sample 10k pseudo-data points for the LLFF dataset. For synthetic and unbounded 360° scenes, we mine 30k samples per scene using Instant-NGP [25] teachers. Following this, we fine-tune the model on the original data. We optimize for the mean-squared error between generated and ground truth images. We refer the reader to the supplementary material for more training details.

We use $63 \times 84$ ($12\times$ downsampled from the desired $756 \times 1008$ resolution) input ray bundles for the forward-facing scenes. For 360° scenes, we use $100 \times 100$ ($8\times$ downsampled from the desired $800 \times 800$ image resolution) ray bundles. For unbounded scenes, we use ray bundles $12\times$ downsampled from the image resolution we use. We train our frontal LightSpeed models as well as each sub-scene model in non-frontal scenes for 200k iterations.

**Baselines and Metrics.** We compare our method's performance on bounded scenes with MobileR2L[6], MobileNeRF[6] and SNeRG[13]. We evaluate our method for rendering quality using three metrics: PSNR, LPIPS, and SSIM. For unbounded scenes, we report the PSNR metric on 6 scenes and compare it with MobileNeRF [6] and NeRFMeshing [27]. To further demonstrate the effectiveness of our approach, we compare our approach with others on two other criteria: (a) **On-device Rendering Speed**: We report and compare average inference times per rendered frame on various mobile chips, including Apple A15, Apple M1 Pro and Snapdragon SM8450 chips; and (b) **Efficient Training**: We compare the number of iterations LightSpeed and MobileR2L require to reach a target PSNR. We pick Lego scene from 360° scenes and Fern from forward-facing scenes as representative scenes to compare. We also report the storage requirements of our method per frontal scene and compare it with baselines.

## 4.1 Results and Analysis

**Rendering Quality.** As in Tab. 1, we obtain better results on all rendering fidelity metrics on the two bounded datasets. We also outperform MobileNeRF and NeRFMeshing on 4 out of 6 unbounded 360° scenes. We refer the reader to Fig. 4 for a visual comparison of our approach with MobileR2L and NeRF. Our method has much better rendering quality, capturing fine-level details where MobileR2L, and in some cases, even the original NeRF model, fails. Note that we use Instant-NGP teachers for 360° scenes, which have slightly inferior performance to MipNeRF teachers used by MobileR2L. This further shows the robustness of our approach to inferior NeRF teachers.

**Storage Cost.** We report storage requirements in Tab. 1. Our approach has a competitive on-device storage to the MobileR2L model. Specifically, we require a total of 16.3 MB of storage per frontal scene. The increase in storage is expected since we're using grids to encode our light field. We also report storage values for lighter LightSpeed networks in the ablation study (see Tab. 5), all of which have similar or better rendering quality than the full-sized MobileR2L network.

**Training Speed.** We benchmark the training times and the number of iterations required for LightSpeed and MobileR2L in Tab. 2 with a target PSNR of 24 for Fern scene and 32 for the Lego scene. Our approach demonstrates a training speed-up of $2.5\times$ on both scenes. Since we are modeling 360° scenes as a composition of 5 light fields, we can train them in parallel (which is not

Table 1: **Quantitative Comparison** on Forward-facing, Synthetic 360° and Unbounded 360° Datasets. LighSpeed achieves the best rendering quality with competitive storage. We use an out-of-the-box Instant-NGP [25] implementation [1] (as teachers for 360° scenes) which dose not report SSIM and LPIPS values. We omit storage for NeRF-based methods since they are not comparable.

| Method | Synthetic 360° | | | Forward-Facing | | | |
|---|---|---|---|---|---|---|---|
| | PSNR ↑ | SSIM ↑ | LPIPS ↓ | PSNR ↑ | SSIM ↑ | LPIPS ↓ | Storage ↓ |
| NeRF [23] | 31.01 | 0.947 | 0.081 | 26.50 | 0.811 | 0.250 | - |
| NeRF-PyTorch | 30.92 | 0.991 | 0.045 | 26.26 | 0.965 | 0.153 | - |
| SNeRG [13] | 30.38 | 0.950 | 0.050 | 25.63 | 0.818 | 0.183 | 337.3 MB |
| MobileNeRF [6] | 30.90 | 0.947 | 0.062 | 25.91 | 0.825 | 0.183 | 201.5 MB |
| MobileR2L [4] | 31.34 | 0.993 | 0.051 | 26.15 | 0.966 | 0.187 | **8.2 MB** |
| LightSpeed (Ours) | **32.23** | **0.994** | **0.038** | **26.50** | **0.968** | **0.173** | 16.3 MB |
| Our Teacher | 32.96 | - | - | 26.85 | 0.827 | 0.226 | - |

| | Unbounded 360° | | | | | |
|---|---|---|---|---|---|---|
| Method | Bicycle | Garden | Stump | Bonsai | Counter | Kitchen |
| MobileNeRF [6] | 21.70 | 23.54 | **23.95** | - | - | - |
| NeRFMeshing [27] | 21.15 | 22.91 | 22.66 | 25.58 | 20.00 | 23.59 |
| LightSpeed (Ours) | **22.51** | **24.54** | 22.22 | **28.24** | 25.46 | **27.82** |
| Instant-NGP (Our teacher) [25] | 21.70 | 23.40 | 23.20 | 27.4 | **25.80** | 27.50 |

Table 2: **Training Time** for Lego and Fern scenes with 32 and 24 target PSNRs. LightSpeed trains significantly faster than MobileR2L. It achieves even greater speedup when trained in parallel for 360° scenes (parallel training is not applicable for frontal scenes).

| | Forward-Facing: Fern | | Synthetic 360°: Lego | |
|---|---|---|---|---|
| Method | Duration ↓ | Iterations ↓ | Duration ↓ | Iterations ↓ |
| MobileR2L | 12.5 hours | 70k | 192 hours | 860k |
| LightSpeed | **4 hours** | **27k** | **75 hours** | **425k** |
| LightSpeed (Parallelized) | - | - | **15 hours** | **85k** |

possible for MobileR2L), further trimming down the training time. Moreover, the training speedup reaches $\sim 4\times$ when networks are trained beyond the mentioned target PSNR (see Fig. 1).

**Inference Speed.** Tab. 3 shows our method's inference time as compared to MobileR2L and MobileNeRF. We maintain a comparable runtime as MobileR2L while having better rendering fidelity. Since on-device inference is crucial to our problem setting, we also report rendering times of a smaller 30-layered decoder network that has similar rendering quality as the MobileR2L model (see Tab. 5).

Table 3: **Rendering Latency Analysis.** LightSpeed maintains a competitive rendering latency (ms) to prior works. MobileNeRF is not able to render 2 out of 8 real-world scenes ($\frac{N}{M}$ in table) due to memory constraints, and no numbers are reported for A13, M1 Pro and Snapdragon chips.

| | Forward-Facing | | | | Synthetic 360° | | | |
|---|---|---|---|---|---|---|---|---|
| Chip | MobileNeRF | MobileR2L | Ours | Ours (30-L) | MobileNeRF | MobileR2L | Ours | Ours (30-L) |
| Apple A13 (Low-end) | - | 40.23 | 41.06 | 32.29 | - | 65.54 | 66.10 | 53.89 |
| Apple A15(Low-end) | 27.15 $\frac{2}{8}$ | 18.04 | 19.05 | 15.28 | 17.54 | 26.21 | 27.10 | 20.15 |
| Apple A15(High-end) | 20.98 $\frac{2}{8}$ | 16.48 | 17.68 | 15.03 | 16.67 | 22.65 | 26.47 | 20.35 |
| Apple M1 Pro | - | 17.65 | 17.08 | 13.86 | - | 27.37 | 27.14 | 20.13 |
| Snapdragon SM8450 | - | 39.14 | 45.65 | 32.89 | - | 40.86 | 41.26 | 33.87 |

## 4.2 Ablations

**Data Requirements.** We use 10k samples as used by MobileR2L to train LightField models for frontal scenes. However, for non-frontal scenes, we resort to using 30k pseudo-data samples per

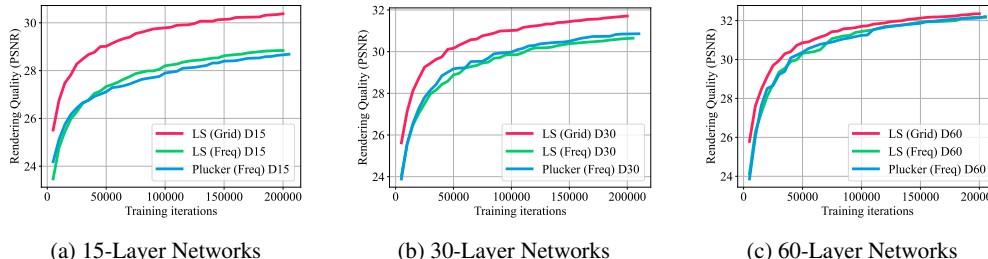

|                | (a) 15-Layer Networks | (b) 30-Layer Networks | (c) 60-Layer Networks |
|----------------|-----------------------|-----------------------|-----------------------|

Figure 5: **Test PSNR v/s Training Iterations.** We compare test set PSNR obtained by LightSpeed (Grid)(ours), LightSpeed (frequency encoded), and Plücker-based neural light field as the training progresses for 3 different network configurations.

scene. Dividing 10k samples amongst 5 sub-scenes assigns too few samplers per sub-scene, which is detrimental to grid learning. We experimentally validate data requirements by comparing MobileR2L and LightSpeed trained for different amounts of pseudo-data. We train one $400 \times 400$ sub-scene from the Lego scene for 200k iterations with 1/5th of 10k and 30k samples, *i.e.*, 2k and 6k samples. Tab. 4 exhibits significantly decreased rendering quality for the LightSpeed network as compared to MobileR2L when provided with less pseudo-data.

Table 4: **Pseudo-Data Requirement for Non-Frontal Scenes.** We analyze the importance of mining more pseudo-data for non-frontal scenes. Using 1/5th of 10k and 30k sampled pseudo-data points, we find more pseudo-data is crucial for the boosted performance of the LightSpeed model.

|                   | 2k Samples |         |           | 6k Samples |         |           |
|-------------------|------------|---------|-----------|------------|---------|-----------|
| Method            | PSNR ↑     | SSIM ↑  | LPIPS ↓   | PSNR ↑     | SSIM ↑  | LPIPS ↓   |
| MobileR2L         | 30.19      | 0.9894  | 0.0354    | 30.56      | 0.9898  | 0.0336    |
| LightSpeed (Ours) | 30.44      | 0.9899  | 0.0299    | **31.2**   | **0.9906** | **0.0284** |

**Decoder Network Size.** We further analyze the trade-off between inference speed and rendering quality of our method and MobileR2L. To this end, we experiment with decoders of different depths and widths. Each network is trained for 200k iterations and benchmarked on an iPhone 13. Tab. 5 shows that a 30-layered LightSpeed model has a better inference speed and rendering quality as compared to the 60-layered MobileR2L model. This 30-layered variant further occupies less storage as compared to its full-sized counterpart. Furthermore, lighter LightSpeed networks obtain a comparable performance as the 60-layered MobileR2L. Note that reducing the network capacity of MobileR2L results in significant drops in performance. This means that we can get the same rendering quality as MobileR2L with considerably reduced on-device resources, paving the way for a much better trade-off between rendering quality and on-device inference speed.

Table 5: **Decoder Network Size.** Our approach maintains a much better tradeoff between inference speeds v/s rendering quality, with our smallest network achieving comparable quality to the MobileR2L. Benchmarking done on an iPhone 13. L is network depth, and W is network width.

| Method               | PSNR ↑ | Latency ↓ | Storage ↓ | FLOPs ↓ |
|----------------------|--------|-----------|-----------|---------|
| 15-L W-256 MobileR2L | 27.69  | 14.54 ms  | 2.4 MB    | 12626M  |
| 30-L W-128 MobileR2L | 27.54  | 14.47 ms  | 1.4 MB    | 8950M   |
| 30-L W-256 MobileR2L | 29.21  | 18.59 ms  | 4.5 MB    | 23112M  |
| 60-L W-256 MobileR2L | 30.34  | 22.65 ms  | 8.2 MB    | 42772M  |
| 15-L W-256 LightSpeed | 30.37 | 14.94 ms  | 10.5 MB   | 12833M  |
| 30-L W-128 LightSpeed | 30.13 | 14.86 ms  | 9.5 MB    | 9065M   |
| 30-L W-256 LightSpeed | 31.70 | 20.35 ms  | 12.6 MB   | 23319M  |
| 60-L W-256 LightSpeed | 32.34 | 26.47 ms  | 16.3 MB   | 42980M  |

**Ray-Space Grid Encoding.** We provide an ablation in Tab. 6 below on how the proposed ray-space grid encoder helps as compared to just using the light-slab representation with a traditional frequency encoder. We compare different LightSpeed configurations with grid-encoder and frequency encoders. Networks are trained for 200k iterations on a full-resolution $800 \times 800$ Lego sub-scene from Synthetic

360° dataset. Further, we show the training dynamics of all the trained variants in Fig. 5 (red and green plots). As claimed, our approach offers better visual fidelity and training dynamics (iterations to reach a target PSNR) for both computationally cheaper small networks as well as full sized networks.

Table 6: **Effect of using a Ray-Space Grid Encoder.** We demonstrate the effect of using a grid-based LightSpeed by comparing with a frequency encoded variant (no grid). L is network depth, and W is network width.

| Method | PSNR ↑ |
|---|---|
| 15-L W-256 LS (PE) | 28.84 |
| 30-L W-256 LS (PE) | 30.63 |
| 60-L W-256 LS (PE) | 32.16 |
| 15-L W-256 LS (Grid) | 30.37 |
| 30-L W-256 LS (Grid) | 31.70 |
| 60-L W-256 LS (Grid) | 32.34 |

**Comparison with Plücker Representation.** Given the challenges of discretizing Plücker representation, we compare between using positionally encoded Plücker coordinates and our grid-based light-slab approach in Tab. 7 below for different network sizes to demonstrate the effectiveness of our approach. We train all models for 200k iterations on one 800×800 Lego sub-scene. We also share training curves for the variants in question in Fig. 5 (red and blue curves). As claimed, our integrated approach performs better in terms of training time and test-time visual fidelity for large and small models (having less computational costs) alike whereas the Plücker-based network shows a sharp decline in visual fidelity and increased training times to reach a target test PSNR as network size is reduced.

Table 7: **Light-Slab Grid Representation vs. Plücker Coordinates.** We compare the light-slab based LightSpeed (LS) with a positionally encoded variant of the Plücker ray representation. L is network depth, and W is network width.

| Method | PSNR ↑ |
|---|---|
| 15-L W-256 Plücker | 28.65 |
| 30-L W-256 Plücker | 30.84 |
| 60-L W-256 Plücker | 32.14 |
| 15-L W-256 LS | 30.37 |
| 30-L W-256 LS | 31.70 |
| 60-L W-256 LS | 32.34 |

## 5 Discussion and Conclusion

In this paper, we propose an efficient method, LightSpeed, to learn neural light fields using the classic two-plane ray representation. Our approach leverages grid-based light field representations to accelerate light field training and boost rendering quality. We demonstrate the advantages of our approach not only on frontal scenes but also on non-frontal scenes by following a divide-and-conquer strategy and modeling them as frontal sub-scenes. Our method achieves SOTA rendering quality amongst prior works at same time providing a significantly better trade-off between rendering fidelity and latency, paving the way for real-time view synthesis on resource-constrained mobile devices.

**Limitations.** While LightSpeed excels at efficiently modeling frontal and 360° light fields, it currently lacks the capability to handle free camera trajectories. The current implementation does not support refocusing, anti-aliasing, and is limited to static scenes without the ability to model deformable objects such as humans. We plan to explore these directions in future work.

**Broader Impact.** Focused on finding efficiencies in novel view synthesis, our study could significantly reduce costs, enabling wider access to this technology. However, potential misuse, like unsolicited impersonations, must be mitigated.

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

## A  Training Details

**Network architecture.**   Our multi-scale feature grids have 16 levels, with resolutions exponentially growing from 16 to 256, and 4-D features in every grid. Our LightSpeed network follows a similar architecture to MobileR2L: 60 point-wise residual convolutions with 256 channels and BatchNorm [15]and GeLU [14] activation interleaved. The convolutions are followed by 3 super-resolution modules to upsample the low-resolution input to the desired resolution. The first two super-resolution modules upsample the input by $2\times$ and consist of transposed convolution layers with $4 \times 4$ kernel size followed by 2 residual convolution layers each. The third super-resolution module consists of transposed kernel size with $4 \times 4$ kernel size (upsample by $2\times$) for $360°$ scenes (both bounded and unbounded) and $3 \times 3$ kernel size (upsample by $3\times$) for forward-facing [22] scenes.

**Training details.**   We use Adam [18] optimizer with a batch size of 32 to train the feature grids and decoder network. We use an initial learning rate of 1e-5 with 100 warmup steps taking the learning rate to 5e-4. Beyond that, the learning rate decays linearly until the training finishes. All our experiments are conducted on Nvidia V100s and A100s.

## B  More Ablation Analysis

**Choice of Splitting Planes.** We discuss two aspects of dividing non-frontal scenes into separate light fields: the number of parts to divide the scene into and the placement of the splitting planes. We find the optimal number of splits for $360°$ scenes to be 5 since more number of splits would mean increased storage cost, which is detrimental to mobile deployment. We also want the scene splits to be collectively exhaustive (but not mutually exclusive to maintain continuity while switching from one light field to another) in the poses sampled around the object. Consequently, fewer planes would mean placing the splitting planes near the scene origin to cover the entire scene, which starts to violate the frontal assumption for each sub-scene.

Given poses distributed on the surface of a sphere with radius $r$, we propose assigning each pose to (possibly multiple) sub-scenes based on the camera origin satisfying one or more of the 5 following criteria:

$$\begin{bmatrix} 0 & 0 & \sqrt{2} \\ \sqrt{2} & 0 & \sqrt{2}-1 \\ -\sqrt{2} & 0 & \sqrt{2}-1 \\ 0 & \sqrt{2} & \sqrt{2}-1 \\ 0 & -\sqrt{2} & \sqrt{2}-1 \end{bmatrix} \begin{bmatrix} x \\ y \\ z \end{bmatrix} \geq \begin{bmatrix} r \\ r \\ r \\ r \\ r \end{bmatrix} \tag{3}$$

These five hyperplanes form the surface of a near-isometric trapezoidal prism, as shown in Fig. 3 (main paper). We experimentally show the effect of the choice of splitting plane by training LightSpeed models on a Lego sub-scene with different plane placements and compare with the corresponding MobileR2L models trained on the same data. Specifically, we choose two axis-aligned planes at a distance of $\frac{radius}{\sqrt{2}}$ and $\frac{radius}{\sqrt{3}}$ from the scene origin and train models with 6k pseudo data points sampled independently from the two resulting sub-scenes. As shown in Tab. 8, placing the splitting plane at a distance of $\frac{radius}{\sqrt{3}}$ results in inferior performance as compared to placing the splitting plane at a distance of $\frac{radius}{\sqrt{2}}$ from the origin. This suggests that frontal sub-scene approximation starts to break down as we move the splitting plane closer to the origin.

Table 8: **Choice of Splitting Planes.** We experiment with two planes parallel to the x-y sub-space at different distances. Splitting planes further from the origin work better empirically maintaining the frontal sub-scene assumption.

| LF Representation | PSNR ↑ | SSIM ↑ | LPIPS ↓ |
|---|---|---|---|
| radius /$\sqrt{2}$ | **30.44** | **0.9903** | **0.028** |
| radius /$\sqrt{3}$ | 30.23 | 0.9899 | 0.031 |

# C Per-Scene Quantitative Results

We provide a per-scene quantitative comparison between LightSpeed, MobileR2L [6] and NeRF [23] on the synthetic 360° dataset (Tab. 9, Tab. 10, and Tab. 11) and forward-facing dataset (Tab. 12, Tab. 13, and Tab. 14). We use PSNR, LPIPS, and SSIM as comparison metrics. As can be seen from the comparisons, LighSpeed (our approach) outperforms MobileR2L [4] on almost all the metrics. Further, LightSpeed performs comparably or even better than NeRF [23]. We also provide additional zoom-in comparisons between LightSpeed and MobileR2L in Fig. 7.

Table 9: Per-scene PSNR ↑ comparison on the Synthetic 360° dataset between NeRF [23], MobileR2L [4], and our approach.

| Method | Chair | Drums | Ficus | Hotdog | Lego | Materials | Mic | Ship | Average |
|---|---|---|---|---|---|---|---|---|---|
| NeRF[23] | 33.00 | 25.01 | 30.13 | 36.18 | 32.54 | 29.62 | 32.91 | 28.65 | 31.01 |
| MobileR2L [4] | 33.66 | 25.05 | 29.80 | 36.84 | 32.18 | 30.54 | 34.37 | 28.75 | 31.34 |
| LightSpeed (Ours) | 34.21 | 25.63 | 32.82 | 36.77 | 34.35 | 29.51 | 35.65 | 28.90 | 32.23 |

Table 10: Per-scene SSIM ↑ comparison on the Synthetic 360° dataset between NeRF [23], MobileR2L [4], and our approach.

| Method | Chair | Drums | Ficus | Hotdog | Lego | Materials | Mic | Ship | Average |
|---|---|---|---|---|---|---|---|---|---|
| NeRF[23] | 0.967 | 0.925 | 0.964 | 0.974 | 0.961 | 0.949 | 0.980 | 0.856 | 0.947 |
| MobileR2L [4] | 0.998 | 0.986 | 0.996 | 0.998 | 0.992 | 0.992 | 0.997 | 0.982 | 0.993 |
| LightSpeed (Ours) | 0.998 | 0.988 | 0.998 | 0.998 | 0.994 | 0.990 | 0.998 | 0.984 | 0.994 |

Table 11: Per-scene LPIPS ↓ comparison on the Synthetic 360° dataset between NeRF [23], MobileR2L [4], and our approach.

| Method | Chair | Drums | Ficus | Hotdog | Lego | Materials | Mic | Ship | Average |
|---|---|---|---|---|---|---|---|---|---|
| NeRF[23] | 0.046 | 0.091 | 0.044 | 0.121 | 0.050 | 0.063 | 0.028 | 0.206 | 0.081 |
| MobileR2L [4] | 0.027 | 0.083 | 0.025 | 0.026 | 0.043 | 0.029 | 0.012 | 0.162 | 0.051 |
| LightSpeed (Ours) | 0.017 | 0.061 | 0.016 | 0.023 | 0.019 | 0.030 | 0.007 | 0.138 | 0.039 |

Table 12: Per-scene PSNR ↑ comparison on the forward-facing dataset between NeRF [23], MobileR2L [4], and our approach.

| Method | Room | Fern | Leaves | Fortress | Orchids | Flower | T-Rex | Horns | Average |
|---|---|---|---|---|---|---|---|---|---|
| NeRF[23] | 32.70 | 25.17 | 20.92 | 31.16 | 20.36 | 27.40 | 26.80 | 27.45 | 26.50 |
| MobileR2L [4] | 32.09 | 24.39 | 20.52 | 30.81 | 20.06 | 27.61 | 26.71 | 27.01 | 26.15 |
| LightSpeed (Ours) | 32.32 | 25.05 | 21.01 | 31.45 | 20.33 | 27.88 | 26.93 | 27.04 | 26.50 |

Table 13: Per-scene SSIM ↑ comparison on the forward-facing dataset between NeRF [23], MobileR2L [4], and our approach.

| Method | Room | Fern | Leaves | Fortress | Orchids | Flower | T-Rex | Horns | Average |
|---|---|---|---|---|---|---|---|---|---|
| NeRF[23] | 0.948 | 0.792 | 0.690 | 0.881 | 0.641 | 0.827 | 0.880 | 0.828 | 0.811 |
| MobileR2L [4] | 0.995 | 0.973 | 0.923 | 0.995 | 0.916 | 0.971 | 0.973 | 0.982 | 0.966 |
| LightSpeed (Ours) | 0.991 | 0.976 | 0.931 | 0.996 | 0.921 | 0.972 | 0.975 | 0.983 | 0.968 |

Table 14: Per-scene LPIPS ↓ comparison on the forward-facing dataset between NeRF [23], MobileR2L [4], and our approach.

| Method | Room | Fern | Leaves | Fortress | Orchids | Flower | T-Rex | Horns | Average |
|---|---|---|---|---|---|---|---|---|---|
| NeRF[23] | 0.178 | 0.280 | 0.316 | 0.171 | 0.321 | 0.219 | 0.249 | 0.268 | 0.250 |
| MobileR2L [4] | 0.088 | 0.239 | 0.280 | 0.103 | 0.296 | 0.150 | 0.121 | 0.217 | 0.187 |
| LightSpeed (Ours) | 0.085 | 0.211 | 0.255 | 0.093 | 0.272 | 0.145 | 0.119 | 0.209 | 0.173 |

## D  Limitations

**Results on Unbounded Scenes.**  The rendering fidelity of LightSpeed is closely tied to the performance of the corresponding NeRF teacher. LightSpeed uses Instant NGP [25] teachers for both bounded and unbounded scenes to maintain experimental consistency. We would like to highlight that Instant-NGP introduces the artifacts to unbounded scenes, which are carried forward to LightSpeed via the mined pseudo-data. We share some of the pseudo-data images from Instant-NGP in Fig. 6. MipNeRF360 [3] specifically uses space contraction techniques to model the unbounded nature of the scene and deal with blurriness in the renderings. It further introduces a distortion-based regularizer to remove floater artifacts and prevent background collapse. The techniques introduced by MipNeRF360 tackle the same type of artifacts pointed out in Fig. 6. Hence, using MipNeRF360 teachers will mitigate both these issues and could boost the visual fidelity on unbounded scenes for LightSpeed.

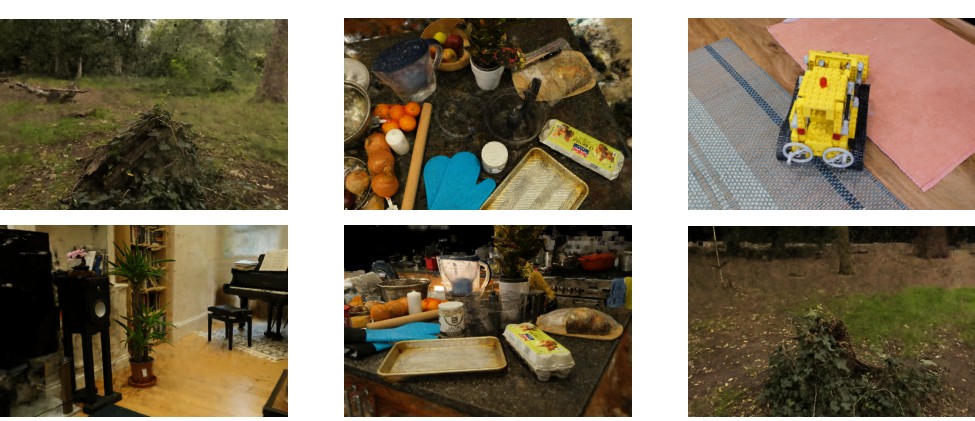

Figure 6: **Instant NGP Failure Cases for Unbounded Scenes.**  Such artifacts carry over to LightSpeed, affecting its visual fidelity on unbounded scenes.

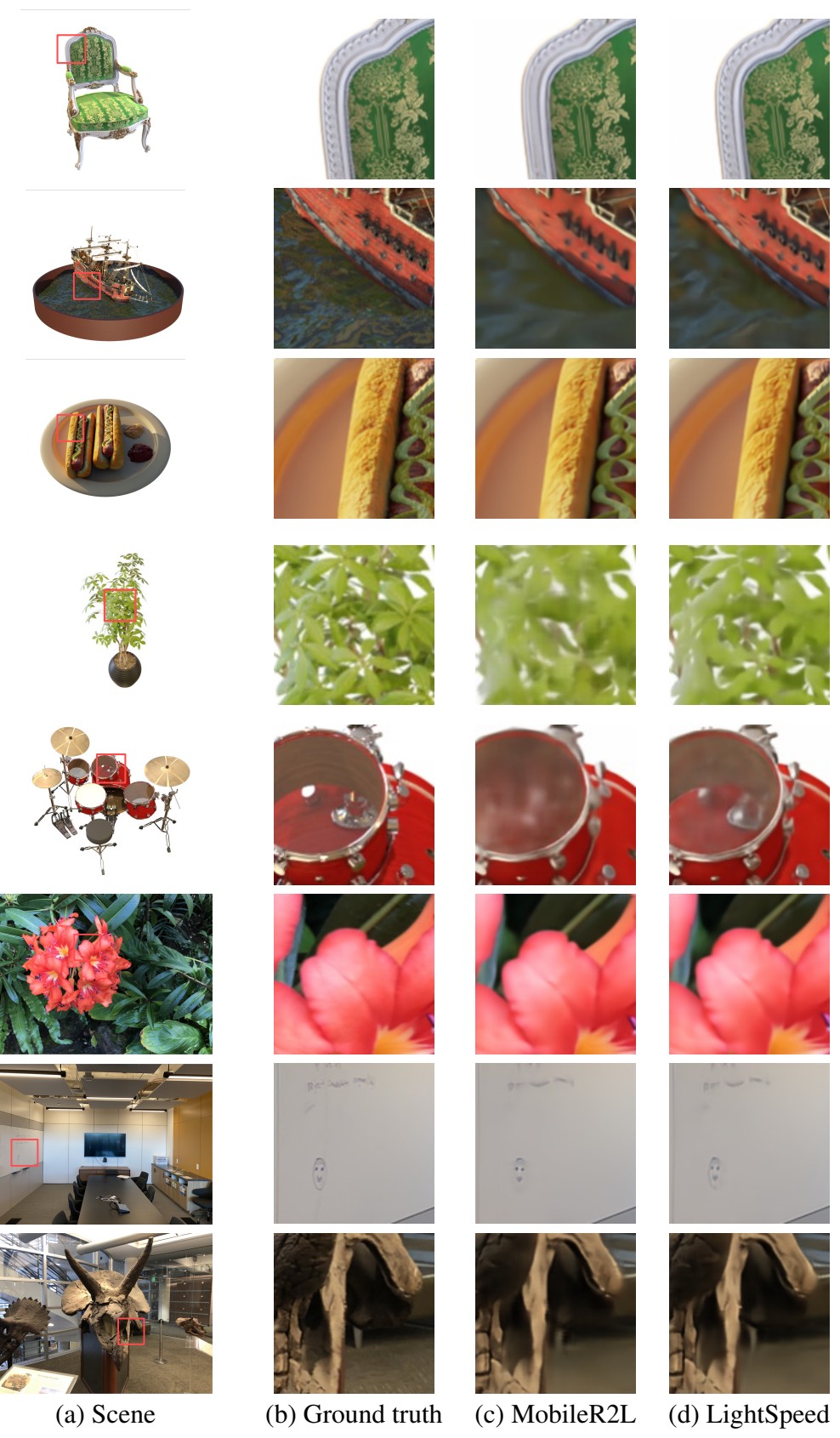

(a) Scene     (b) Ground truth     (c) MobileR2L     (d) LightSpeed

Figure 7: **Qualitative Results on frontal and non-frontal scenes.** Zoomed-in comparison between MobileR2L and our LightSpeed approach.

