# OpenReview forum: "LightSpeed: Light and Fast Neural Light Fields on Mobile Devices"
_NeurIPS.cc/2023/Conference — NeurIPS 2023 poster_

### Official Review · Reviewer_CSSm · 2023-06-29

**Soundness:** 4 excellent
**Presentation:** 4 excellent
**Contribution:** 3 good
**Rating:** 6
**Confidence:** 4

**Summary:**

This paper describes a novel representation for learning view synthesis from a set of input images with known camera poses.  They parameterize a classical two-slab 4D light field using a K-Planes representation (using 6 feature planes).  Feature queries are processed through many layers of 1x1 convolutions before being post-processed by a super-resolution network to produce the output image.  Similar to previous neural light field methods they augment the training data using virtual views rendered using a trained NeRF model.  In experiments on standard benchmark datasets they achieve good rendering quality compared to previous methods capable of rendering on mobile devices, and also achieve a fast training time and compact representation.

**Strengths:**

The proposed approach of parameterizing a neural lightfield using the K-Planes concept is reasonable and novel to the best of my knowledge.  They show convincingly that this approach leads to faster convergence and a more compact representation compared to MobileR2L.

They also achieve high-quality rendering with fast rendering speeds even on mobile devices.  In some cases they actually outperform the teacher model.

They present an extensive evaluation using several benchmark datasets, including both synthetic and real data.  They also include an ablation study to consider the effect of virtual views and decoder network size.  They also compare implementations on different mobile processors in terms of latency.

The presentation is clear and easily understandable.

**Weaknesses:**

The results on unbounded scenes especially are lower quality than state-of-the-art (non-real-time) methods such as Mip-NeRF 360.  Artifacts such as blurriness and inconsistent shape are clearly evident in the result videos for unbounded scenes (but not for the object-centric scenes).  These limitations deserve discussion.

The approach of rendering a low-resolution image and then upsampling would seem to be a limiting factor in terms of rendering quality.  It would be informative to see what quality is possible with this representation when directly rendering the full-resolution image (without the super-resolution network).

**Questions:**

How do the results compare to state-of-the-art (regardless of rendering speed)?

What is the effect of directly rendering a full-resolution image (rather than using the super-resolution network)?

**Limitations:**

They do describe limitations but I think more discussion of the 360 unbounded results would strengthen the paper.

---

> ### Author Rebuttal · Authors · 2023-08-09
>
>
> **We thank the reviewer for their positive comments and insightful feedback. We appreciate that the reviwer acknowledges our approach to be novel with fast and high-quality renderings. We further note the reviewer finds evaluations extensive and the paper's presentation clear and easily understandable. In the following, we address the feedback and questions presented by the reviewer.**
>
> ***
> **Q1. Regressing full-resolution image directly.**
>
> We thank the reviewer for raising an insightful point! In our preliminary experiments, we found a *slight* increase in visual fidelity if we regress the full-resolution image directly. However, since rendering full-resolution images directly is not feasible on mobile devices in real-time (as pointed out in MobileR2L [a]) and the visual fidelity gains are marginal, we did not pursue this direction of experiments.
>
>
> ***
> **Q2. About more comparisons.**
>
> Our method is tailored specifically for mobile devices, and achieves state-of-the-art rendering fidelity on both LLFF and Synthetic $360^\circ$ scenes compared to prior methods in this domain (Tab. G below). Given the use of teacher NeRF methods to generate pseudo-data for light field training, we can potentially improve the rendering fidelity of LightSpeed by leveraging newer NeRF-based methods.
>
>
> >Table G: **Quantitative Comparison** on Forward Facing and Synthetic $360^\circ$ scenes.
> >| Method  | Synthetic $360^\circ$ PSNR $\uparrow$| LLFF PSNR $\uparrow$ |
> >| :------------ | :-----------: | :-----------: |
> >| NeRF               | 31.01 |26.50|
> >| NeRF-PyTorch       | 30.92 |26.26|
> >| SNeRG              | 30.38 |25.63|
> >| MobileNeRF         | 30.90 |25.91|
> >| MobileR2L          | 31.34 |26.15|
> >| LightSpeed (Ours)  | **32.23** | **26.50**|
>
>
> ***
> **References:**
>
> [a] Cao, Junli, et al. "Real-Time Neural Light Field on Mobile Devices." CVPR. 2023.

---

> > ### Comment · Reviewer_CSSm · 2023-08-16
> >
> > Thanks to the authors for their responses.  I have read over the other reviews and the authors' responses.  I think they have sufficiently addressed the concerns raised in the reviews and I will maintain my recommendation of acceptance.

---

> > > ### Author Response · Authors · 2023-08-17
> > >
> > > Thank you so much for checking our response and other reviews as well. We appreciate your time and reviewing efforts. If you have any other questions or concerns, let us know and we will make the best of our efforts to resolve them within the open discussion period.
> > >
> > > Best,
> > >
> > > Authors

---

### Official Review · Reviewer_8eAH · 2023-07-01

**Soundness:** 2 fair
**Presentation:** 3 good
**Contribution:** 3 good
**Rating:** 6
**Confidence:** 3

**Summary:**

This paper presents the LightSpeed for real-time rendering on mobile devices. The approach involves replacing the commonly used Plücker coordinates with a light slab representation and implementing multi-level grids similar to Instant-NGP and k-planes. Additionally, It introduced a divide-and-conquer strategy to address the issue of light slab's inability to represent 360-degree objects effectively.

**Strengths:**

The proposed approach effectively improves the efficiency with several interesting design. Firstly, it uses light-slab to parameterize ray, which shows efficiency compared to the commonly used Plücker coordinate.  Besides, the practice of compressing 4D space through six sub-planes can also improve efficiency.  Finally, the divide-and-conquer strategy to solve the problem that traditional light-slab method fail to represent 360 degree objects.

**Weaknesses:**

1. In the abstract and introduction sections, the authors repeatedly claim that existing methods overlook the light-slab parameterization. However, research has already been conducted on novel view synthesis based on light-slab parameterization. Besides Attal et al.  mentioned in line 69, there are some other literatures that are related but ignored, such as
NeuLF: Efficient Novel View Synthesis with Neural 4D Light Field
NeLF: Practical Novel View Synthesis with Neural Light Field
Progressively-connected Light Field Network for Efficient View Synthesist
All these papers all based on light-slab parametrization and its variants, and thus should be carefully discussed or compared with.

2. Some claims are hard to understand. For example, in lines 51-52, current ray parametrization will encounter issues with the introduction of grid-based representations. However, no specific problem is identified, and the subsequent description only explains that the existing method is "redundant." Also, in line 57,  “the high-dimensional stratified-point representation is not feasible for grid-based discretization.”, it is unclear why it was not feasible. In my understanding, nerf is also a neural representation that needs sampling, and it can use the grid-based representation.

3. In the ablation section, the authors only ablate the data requirements and the decoder network size, which is not sufficient. More in-depth discussions are needed to demonstrate the effectiveness of each part of the proposed method. For example, it would be great if the authors could make comparison between the grid-based approach and the pure MLP approach within their own framework, such as replacing the Ray-space Grid Encoder with a traditional frequency encoder or so on. Please carefully check the claimed contributions and ensure that all of them are supported by the experiments.

**Questions:**

Please address my questions in the weakness section.

**Limitations:**

Yes

---

> ### Author Rebuttal · Authors · 2023-08-09
>
>
> **We thank the reviewer for their positive review and feedback. We appreciate that the reviewer acknowledges the efficiency of our method and interesting design choices of using a light-slab ray paramterization with 4D grid compression via decomposition. We address the feedback provided by the reviewer in the following.**
>
> ***
> **Q1. About the compatibility of other ray parameterizations with grid-based representations, L51-52, 57.**
>
> **(a). Plücker representation.**
> Plücker representation lies in the projective 5D space, presenting challenges for discretization and grid-based learning. Even if we ignore the projective nature, discretization results in a 5D ray space which (in both original and decomposed form) has a bigger storage cost as compared to its light-slab counterpart. Given the target devices are mobile, storage must be as limited as possible.
>
> **(b). Stratified ray representation.**
> Further, the stratified ray representation used by R2L [a] and MobileR2L [b] can be potentially discretized in two ways: (1) defining three new dimensions for every point sampled along the ray, which results in an extremely overparameterized and *redundant* ray space with unusable storage costs, and (2) querying a spatial 3D grid for points sampled along the ray. Multiple queries of the 3D grid per ray as done by NeRF-based counterparts *increases run-time per pixel*, prohibiting real-time inference on mobile devices. Alternatively, the light-slab ray-space grid is compact and allows a *single grid query per ray/pixel* conducive to real-time inference. Hence, we find light-slab paramterization to be most effective and others presenting issues towards grid-based learning.
>
> ***
> **Q2. Ablation on ray-space grid encoding.**
>
> We provide an ablation in Tab. E below on how the proposed Ray-Space Grid Encoder helps as compared to just using the light-slab representation with a traditional frequency encoder. For the purpose of this ablation, we train LightSpeed with grid-encoder and frequency encoders for 200k iterations with different network sizes and compare results on a full-resolution 800X800 Lego sub-scene from Synthetic $360^\circ$ dataset. Further, we show the training dynamics for all the trained variants in Fig. 2 of rebuttal PDF (red and green plots). As claimed, our approach offers better visual fidelity and training dynamics (iterations to reach a target PSNR) for both computationlly cheaper small networks as well as full sized networks.
>
>
> >Table E: **Effect of using a Grid Encoder**: We demonstrate the effect of using a grid-based LightSpeed by comparing with a frequency encoded variant (no grid). L and W refer to network depth and width respectively.
> >| Method  | PSNR $\uparrow$|
> >| :------------ | :-----------: |
> >| 15-L W-256 LS (PE) | 28.84          |
> >| 30-L W-256 LS (PE) | 30.63          |
> >| 60-L W-256 LS (PE) | 32.16          |
> >| 15-L W-256 LS (Grid) |30.37 |
> >| 30-L W-256 LS (Grid) |31.70 |
> >| 60-L W-256 LS (Grid) | 32.34|
>
> ***
> **Q3. Full-resolution ablation.**
>
> We show visual fidelity and on-device latency tradeoff at *full-resolution* in Tab. F below. We also report FLOP values as an indicator of computational resources required at run-time. LightSpeed maintains a significantly better tradeoff as compared to MobileR2L on full resolutions scenes as well.
>
>
> >Table F: **Full-Resolution Fidelity-Latency Tradeoff**: LightSpeed (LS) maintains a much better tradeoff than MobileR2L (MR2L). Benchmarking done on an iPhone 13 with full-resolution images. L is network depth, and W is network width.
> >| Method  |  PSNR $\uparrow$| Latency $\downarrow$| FLOPs $\downarrow$|
> >| :------------ | :-----------: | :-----------: | :-----------: |
> >| 15-L W-256 MR2L | 27.69 | 14.54 ms | 12626M |
> >| 30-L W-128 MR2L | 27.54 | 14.47 ms | 8950M |
> >| 30-L W-256 MR2L | 29.21 | 18.59 ms | 23112M |
> >| 60-L W-256 MR2L |30.34 | 22.65 ms | 42772M |
> >| 15-L W-256 LS | 30.37 | 14.94 ms | 12833M |
> >| 30-L W-128 LS | 30.13 | 14.86 ms | 9065M |
> >| 30-L W-256 LS | 31.70 | 20.35 ms | 23319M |
> >| 60-L W-256 LS | 32.34 | 26.47 ms | 42980M |
>
>
>
> ***
> **References:**
>
> [a] Wang, Huan, et al. "R2l: Distilling neural radiance field to neural light field for efficient novel view synthesis." ECCV. 2022.
>
> [b] Cao, Junli, et al. "Real-Time Neural Light Field on Mobile Devices." CVPR. 2023.

---

> ### Author Response · Authors · 2023-08-15
>
> Dear Reviewer 8eAH,
>
> We sincerely thank you again for your thoughtful suggestions and valuable feedback to improve our work.
>
> We provide additional explanations to help clarify our work. As the deadline for open discussion is soon, we sincerely hope to use this opportunity to see if our responses are sufficient and if any concern remains. It would be our great pleasure if you would consider updating your review or score.
>
> Thanks again for your time.
>
> Best,
>
> Authors

---

> > ### Comment · Reviewer_8eAH · 2023-08-15
> >
> > Thanks for providing the response! All my concerns are addressed. And thus I would like to raise my rating to "weak accept.

---

> > > ### Author Response · Authors · 2023-08-16
> > >
> > > Dear Reviewer 8eAH,
> > >
> > > Thank you so much for checking our responses and raising the score. It is our great pleasure to know our efforts have helped address your concerns!
> > >
> > > We appreciate your time and reviewing efforts to help improve our work. If you still have questions or concerns, we would sincerely like to know and will make the best of our efforts to resolve them within the open discussion period.
> > >
> > > Best,
> > >
> > > Authors

---

### Official Review · Reviewer_BeXQ · 2023-07-06

**Soundness:** 3 good
**Presentation:** 2 fair
**Contribution:** 2 fair
**Rating:** 5
**Confidence:** 4

**Summary:**

This paper introduces LightSpeed, which uses traditional 4D light-slab representation and merges the super-resolution network proposed by MobileR2L. LightSpeed uses the NeLF method and will be primarily implemented on mobile.

**Strengths:**

Originality: Utilize the overlooked method of 4D light-slab representation and merge the good method of saving memory and time from other NeLF methods (The super-resolution network proposed by MobileR2L). Extend the application scenes of light-slab representation to non-frontal scenes using the divide-and-conquer strategy.

Quality: Greatly saves the storage compared with other methods and balances the reconstruction quality and storage well.

Clarity: Clearly show the results and the advantages of LightSpeed.

Significance: Advance the general application of the NeLF method on mobile.


**Weaknesses:**

Since LightSpeed was proposed to solve the reconstruction problem in mobile, this paper does not show enough data for different mobile platforms. This paper will be more convincing with more data in different chips or multi-platform.

This paper only shows the examples of unbounded datasets in Fig.4, and more comparisons of other datasets should be shown to be more convincing. The data demonstrated now keeps me in doubt about the effects.

Table 3 should be placed in section 4.2 rather than 4.3.


**Questions:**

I think the storage problem is not the most significant in the rendering problem in mobile. The mobile device can upload data to the cloud to solve this problem. RealityScan adopts this method and can obtain similar results based on NeRF method. So what is the most significant strength of LightSpeed for a user who can use cloud storage? Moreover, since MobileNeRF can achieve real-time manipulation, can LightSpeed achieve similar effects? Besides, demonstrating more data will make this paper more convincing, like experimenting on more different chips.

**Limitations:**

please see weaknesses.

---

> ### Author Rebuttal · Authors · 2023-08-09
>
> **We thank the reviewer for their positive feedback and valuable suggestions. We appreciate that the reviewer finds our work original, explained clearly and of significance towards light field methods for mobile devices. We address concerns via more visual and on-device results and hope our response demonstrates the strengths of our approach.**
>
> ***
> **Q1. Latency on different chipsets.**
>
> We compare the latency of our approach with MobileR2L on 6 different chipsets as shown in Tabs. D-1 and D-2 below. We obtain competitive latency numbers for our full-sized LightSpeed network, and much better latency for our 30-layered network (on all devices), which has better rendering fidelity than full-sized MobileR2L as shown in Tab. D-3 below.
>
>
> > Table D-1: **Rendering Latency Analysis on LLFF Scenes**: LightSpeed maintains a competitive rendering latency (ms) to prior works.
> >| Chip  |  MobileR2L| Ours | Ours (30-L) |
> >| :------------ | :-----------: | :-----------: | :-----------: |
> >| Apple A13 (Low-end)  | 40.23 | 41.06 |  32.29|
> >| Apple A15(Low-end)  | 18.04 |  19.05 | 15.28|
> >| Apple A15(High-end) |  16.48 | 17.68 | 15.03|
> >| Apple A16  | 17.84 | 18.15 |  14.83 |
> >| Apple M1 Pro  | 17.65 | 17.08 | 13.86 |
> >| Snapdragon SM8450  | 39.14 | 45.65 | 32.89|
>
>
> >Table D-2: **Rendering Latency Analysis on Synthetic $360^\circ$ Scenes**: LightSpeed maintains a competitive rendering latency (ms) to prior works.
> >| Chip  |  MobileR2L| Ours | Ours (30-L) |
> >| :------------ | :-----------: | :-----------: | :-----------: |
> >| Apple A13 (Low-end)  | 65.54 | 66.10 |  53.89|
> >| Apple A15(Low-end)  | 26.21 | 27.10|  20.15 |
> >| Apple A15(High-end) |  22.65 | 26.47 | 20.35|
> >| Apple A16  | 25.98 | 26.44 |  20.46 |
> >| Apple M1 Pro  | 27.37 | 27.14 | 20.13 |
> >| Snapdragon SM8450  | 40.86 | 41.26 | 33.87|
>
>
>
> >Table D-3: **Full-Resolution Fidelity-Latency Tradeoff**: LightSpeed (LS) maintains a much better tradeoff than MobileR2L (MR2L). Benchmarking done on an iPhone 13 with full-resolution images. L is network depth, and W is network width.
> >| Method  |  PSNR $\uparrow$| Latency $\downarrow$| FLOPs $\downarrow$|
> >| :------------ | :-----------: | :-----------: | :-----------: |
> >| 15-L W-256 MR2L | 27.69 | 14.54 ms | 12626M |
> >| 30-L W-128 MR2L | 27.54 | 14.47 ms | 8950M |
> >| 30-L W-256 MR2L | 29.21 | 18.59 ms | 23112M |
> >| 60-L W-256 MR2L |30.34 | 22.65 ms | 42772M |
> >| 15-L W-256 LS | 30.37 | 14.94 ms | 12833M |
> >| 30-L W-128 LS | 30.13 | 14.86 ms | 9065M |
> >| 30-L W-256 LS | 31.70 | 20.35 ms | 23319M |
> >| 60-L W-256 LS | 32.34 | 26.47 ms | 42980M |
>
> ***
>
> **Q2. Visual results.**
>
> We show all video results in the supplementary material and comparison results on 4 different scenes in Fig. 4 and Fig. 1 of the main paper and supplementary material respectively. We further share results on more scenes in Fig. 1 of the rebuttal PDF to strenghten our claims. As claimed, LightSpeed is able to capture fine-level details better than previous state-of-the-art MobileR2L [a]. Competitive on-device runtimes and significantly better visual fidelity-latency tradeoff further demonstrate the strengths of our work.
>
> ***
> **Q3. About the placement of the table.**
>
> Thank you for pointing this out. We will make sure Table 3 is placed in Sec. 4.2 in the revised paper.
>
> ***
> **Q4. Using cloud storage.**
>
> Uploading LightSpeed models to the cloud would add additional latency to the test-time view-synthesis process, destroying the purpose of using a real-time rendering framework like LightSpeed. To use cloud storage for unnloading models off the mobile devices, we would still want the models to be small enough for fast on-loading and off-loading (bigger models would introduce more overhead). Additionally, privacy concerns and monetary costs also arise if models are stored in the cloud.
>
> ***
> **Q5. Real-time scene manipulation.**
>
> Since light fields inherently do not model the scene geometry, scene manipulation is not currently possible with LightSpeed. There are possibilities for composing two light fields for scene manipulation, and we plan to explore this in future work. MobileNeRF uses explicit scene representation in the form of a mesh and hence can manipulate scenes.  However, MobileNeRF does not run on mobile devices for all scenes: it runs out of memory for complex scenes and hence presents a crucial drawback by not supporting different kinds of scenes. On the contrary, LightSpeed does not suffer from any such drawbacks and can easily handle complex scenes as well within on-device computational limits.
>
> ***
> **References:**
>
> [a] Cao, Junli, et al. "Real-Time Neural Light Field on Mobile Devices." CVPR. 2023.

---

> > ### Comment · Reviewer_BeXQ · 2023-08-20
> >
> > Thanks for addressing our concerns. We will keep our original rating for this paper.

---

### Official Review · Reviewer_XtjZ · 2023-07-07

**Soundness:** 3 good
**Presentation:** 3 good
**Contribution:** 3 good
**Rating:** 8
**Confidence:** 2

**Summary:**

Real-time novel-view image synthesis on mobile devices is challenging due to limited computational power and storage. Volumetric rendering methods are unsuitable due to their high computational cost. The authors propose using the efficient light slab representation for learning a neural light field, which achieves better rendering quality and a favorable trade-off between quality and speed compared to prior light field methods.

**Strengths:**

The paper proposes to use the light slab representation for learning a neural light field, which has not been used significantly in the literature before. The proposed method using light slab presentation is shown to perform better than the SOTA while providing a computational advantage over the existing methods.

**Weaknesses:**

NA

**Questions:**

N/A

**Limitations:**

Yes

---

> ### Author Rebuttal · Authors · 2023-08-09
>
>
> **We thank the reviewer for their feedback and strong rating. As summarized in the review, we propose a novel real-time view-synthesis method that is based on light fields. We leverage previously overlooked 4D light-slab representation for easy discretization and grid-based representations for neural light field learning. Our grid-based neural light field obtains a significant boost in training speed and performs better than exsiting works. Our approach further provides a computational advantage over existing methods in the form of an excellent tradeoff between rendering fidelity and on-device latency, paving the way for easy deployment to mobile devices.**

---

### Official Review · Reviewer_bQpx · 2023-07-12

**Soundness:** 2 fair
**Presentation:** 3 good
**Contribution:** 2 fair
**Rating:** 4
**Confidence:** 5

**Summary:**

The paper introduces LightSpeed, a method aimed at simplifying real-time novel-view image synthesis on mobile devices, which typically face constraints related to computational power and storage. By adopting the traditionally underutilized 4D light-slab (two-plane) representation for learning a neural light field, LightSpeed offers a more compact and efficient ray representation. While the light-slab representation has its limitations, mainly being designed for frontal views, the paper demonstrates a way to extend it for non-frontal scenes using a divide-and-conquer strategy.

**Strengths:**

- The paper proposes a promising approach that integrates neural light field representation with grid representation.

**Weaknesses:**

- The Contribution Context: While the methodology primarily integrates existing techniques like the neural light field method and grid representation from k-plane[9] and tensorf[5], it lacks a direct comparison or detailed discussion on the light-slab representation versus the Plücker coordinate representation[26] for non-frontal scenes. Given that the Plücker coordinate representation might achieve similar results, an experimental comparison would provide a more definitive understanding of the actual performance improvements, if any, achieved by their proposed method. Although such an integrated approach has its merits, the overall performance improvement appears to be marginal without these comparative analyses.

- Coverage of Related Work: The paper mentions another neural light field method using light-slab representation. However, other related and potentially influential works such as "Neulf: Efficient novel view synthesis with neural 4D light field, EGSR 2022" and "Signet: Efficient neural representation for light fields, ICCV 2021" have been overlooked.

- Limited Results: The experimental results presented have some limitations. Notably, the occurrence of the 'jelly effect', particularly in unbounded scenes, implies that the proposed method could benefit from further optimization. Furthermore, the synthetic scene experiments were conducted at a resolution of 400 x 400, as highlighted in Figure 1 and the ablation study. It remains uncertain how the proposed method would perform at the dataset's original resolution of 800 x 800. This higher-resolution evaluation could provide a more thorough understanding of the method's capabilities.

- Feasibility for Real-Time Rendering: The paper claims real-time rendering feasibility on mobile devices, but it does not provide sufficient evidence such as measured MACs or a real-time rendering video demonstration. The computational cost of the 30-layered decoder plus 6-plane feature query might prove too expensive for the intended mobile applications. Further investigations should be conducted to substantiate these claims.

**Questions:**

-In line 175, it is unclear how the authors decided on the locations for the two planes, P1 and P2. Could the authors provide clarification on their choice of plane locations?

-It is also not mentioned whether positional encoding was utilized before feeding data into the network. Could the authors specify if this step was incorporated in their method?

**Limitations:**

The paper presents a noteworthy approach by merging neural light field representation with grid representation. However, it primarily rehashes existing methods, with limited novel contribution. Experimental results exhibit limitations, and concerns about the feasibility of real-time rendering on mobile devices persist. Therefore, considering these constraints, I suggest a borderline reject for this paper in its current state.

---

> ### Author Rebuttal · Authors · 2023-08-09
>
>
> **We thank ther reviewer for the valuable feedback. We appreciate that the reviewer finds our approach of  integrating light fields with grid-based representations noteworty and promising. We address the concerns in the following. We hope our response can further demonstrate the strengths and real-time feasibility of our method.**
>
>
> ***
> **Q1. About issues for Plücker representation discretization.**
>
> Plücker representation lies in the projective 5D space, presenting challenges for discretization and grid-based learning. Even if we ignore the projective nature, discretization results in a 5D ray space which (in both original and decomposed form) has a bigger storage cost as compared to its light-slab counterpart. Given the target devices are mobile, storage shall be as limited as possible.
>
> ***
> **Q2. Comparison with Plücker representation.**
>
> Given the challenges of discretizing Plücker representation, we show a comparison between using positionally encoded Plücker coordinates and our grid-based light-slab approach in Tab. A below for different network sizes to demonstrate the effectiveness of our approach. We train all models for 200k iterations on one Lego sub-scene at the *full 800x800 resolution*. We also share training curves for the variants in question in Fig. 2 of rebuttal PDF (red and blue curves). As claimed, our integrated approach performs better in terms of training time and test-time visual fidelity for large and small models (having less computational costs) alike whereas the Plücker-based network shows a sharp decline in visual fidelity and increased training times to reach a target test PSNR as network size is reduced.
>
> >Table A: **Light-Slab Grid Representation vs. Plücker Coordinates:** We compare the light-slab based LightSpeed (LS)  with a positionally encoded variant of the Plücker ray representation.
> >| Method  | PSNR $\uparrow$|
> >| :------------ | :-----------: |
> >| 15-L W-256 Plücker | 28.65  |
> >| 30-L W-256 Plücker | 30.84  |
> >| 60-L W-256 Plücker | 32.14  |
> >| 15-L W-256 LS | 30.37 |
> >| 30-L W-256 LS | 31.70 |
> >| 60-L W-256 LS | 32.34|
> >
>
> ***
> **Q3. Full-resolution ablation.**
>
> Our evaluations in Tab. 1 of the main paper (cropped version as Tab. B below) are conducted at *full resolution*. We further show visual fidelity and on-device latency tradeoff at full-resolution in Tab. C below. LightSpeed maintains a significantly better tradeoff as compared to MobileR2L on full resolutions scenes as well.
>
>
>
> >Table B: **Quantitative Comparison**: on Forward Facing and Synthetic $360^\circ$ scenes.
> >| Method  | Synthetic $360^\circ$ PSNR $\uparrow$| LLFF PSNR $\uparrow$ |
> >| :------------ | :-----------: | :-----------: |
> >| NeRF               | 31.01 |26.50|
> >| NeRF-PyTorch       | 30.92 |26.26|
> >| SNeRG              | 30.38 |25.63|
> >| MobileNeRF         | 30.90 |25.91|
> >| MobileR2L          | 31.34 |26.15|
> >| LightSpeed (Ours)  | **32.23** | **26.50**|
>
>
> >Table C: **Full-Resolution Fidelity-Latency Tradeoff**: LightSpeed (LS) maintains a much better tradeoff than MobileR2L (MR2L). Benchmarking done on an iPhone 13 with full-resolution images. L is network depth, and W is network width.
> >| Method  |  PSNR $\uparrow$| Latency $\downarrow$| FLOPs $\downarrow$|
> >| :------------ | :-----------: | :-----------: | :-----------: |
> >| 15-L W-256 MR2L | 27.69 | 14.54 ms | 12626M |
> >| 30-L W-128 MR2L | 27.54 | 14.47 ms | 8950M |
> >| 30-L W-256 MR2L | 29.21 | 18.59 ms | 23112M |
> >| 60-L W-256 MR2L |30.34 | 22.65 ms | 42772M |
> >| 15-L W-256 LS | 30.37 | 14.94 ms | 12833M |
> >| 30-L W-128 LS | 30.13 | 14.86 ms | 9065M |
> >| 30-L W-256 LS | 31.70 | 20.35 ms | 23319M |
> >| 60-L W-256 LS | 32.34 | 26.47 ms | 42980M |
>
>
> ***
> **Q4. On-device feasibility.**
>
> We report the FLOPs for our method and MobileR2L in Tab. C. We demonstrate *real-time latency numbers* obtained from benchmarking the LightSpeed framework *directly on mobile devices* using the Xcode from Apple. Real-time demo to be released with the code later. Specifically, using a 30-layered decoder plus a six-plane feature query has almost half the operations of full-sized network from MobileR2L, both of which run on mobile devices.
>
> ***
> **Q5. Location for planes P1 and P2.**
>
> We use the same NDC trick as leveraged by NeRF to project rays to the NDC space: project the scene's near plane to z = -1 and the plane at infinity to z = 1. We use these projected planes (z=+-1) as planes P1 and P2.
>
> ***
> **Q6. Use of positional encoding.**
>
> *No positional encoding* is utilized before feeding the grid encodings to the decoder network. Grid-based representations and positional encodings offer alternative ways to provide interpolation capabilities to the network, and hence, using one of them is sufficient.

---

> > ### Comment · Reviewer_bQpx · 2023-08-18
> >
> > Thank you to the authors for the detailed rebuttal and additional experiments provided. The response has indeed addressed some of my initial concerns, though I still have noteworthy reservations about the paper.
> >
> > - **Light Slab Representation**: As also highlighted by Reviewer 8eAH, this parametrization has been explored in several pieces of literature. Please ensure you verify if the references are from arXiv preprints or are already published. A revision of the claims and a detailed discussion comparing to these methods is necessary.
> >
> > - **Regarding the Plucker representation**: The authors have responded by suggesting that a 5d space occupies more space than a 4d one. However, in the context of a 360-degree scene, the 4d representation would necessitate multiple light slab representations, which could also be space-consuming. While I appreciate the inclusion of a table that quantitatively compares the light slab representation, demonstrating its superiority, I feel that the advantage might be marginal.
> >
> > - **Concerning the real-time mobile rendering claim**: The title **"lightSpeed"** suggests an extremely efficient and low power consumption solution. However, according to the rebuttal, the configuration "60-L W-256 LS", which surpasses the quality of mobileR2L, achieves just under 40 fps on the iPhone 13 with an A15 chip. This might be even more challenging for Android devices with lesser computational capabilities. Moreover, the authors have mentioned providing a real-time demo only post the code release. Given the bold "lightSpeed" claim and considering NeurIPS's stature, I believe a video demo should be available for review prior to the review process conclusion. Could the authors provide a live demonstration video through an anonymous link?
> >
> > In conclusion, I believe the paper presents an interesting contribution. However, I'd urge the authors to revisit some of their claims and experimental evidence. My reservations, especially concerning the real-time rendering on mobile devices, remain. I am open to reconsidering my evaluation if further convincing evidence is presented.

---

> ### Author Response · Authors · 2023-08-15
>
> Dear Reviewer bQpx,
>
> We sincerely thank you again for your thoughtful suggestions and valuable feedback to improve our work.
>
> We provide additional explanations to help clarify our work. As the deadline for open discussion is soon, we sincerely hope to use this opportunity to see if our responses are sufficient and if any concern remains. It would be our great pleasure if you would consider updating your review or score.
>
> Thanks again for your time.
>
> Best,
>
> Authors

---

> ### Author Response · Authors · 2023-08-21
>
> Thank you for taking the time to check our response. We try our best to clarify the reservations you may have about the paper.
>
> ***
> **Q1. Light Slab Representation**
>
> We provide a detailed discussion of the related works pointed out by reviewer and Reviewer `8eAH` in Q1 of the common response to all the reviews (`Author Rebuttal by Authors`). We will ensure that a discussion is added to the revised paper along with proper references. We humbly think we have addressed the concerns for Reviewer `8eAH` in this regard.
>
> ***
>
> **Q2. Regarding the Plucker representation**
>
> We appreciate the reviewer's questions about the efficiency of a grid-based discretized Plucker method over our light-slab based method. We experimentally try to discretize the 5D Plucker space and model the ray-space using $5 \choose 2$ 2D feature grids using our framework. All our efforts towards this approach fail. The main issue we encounter is that removing the projective ambiguity requires that we fix one of the coordinates to a constant value. However, we encounter *degenerate cases* where the coordinate to be normalized becomes 0. The best performance with this approach is compared with our LightSpeed model in Tab. A. These experiments are conducted for 200k iterations on a Lego scene. The grid-based Plucker representation isn’t able to learn anything compared to our method.
>
> > Table A: **Grid-based Plucker Representation**: Plucker-grid representation fails to learn the scene compared to our method.
> >| Method |  PSNR $\uparrow$ |
> >| :------------ | :-----------: |
> >| Plucker-Grid  | 13.36 |
> >| Ours  | 31.20 |
>
> We hypothesize that this poor performance stems from the projective nature of the Plucker representation that *hinders* the effective discretization of the corresponding ray space and hence grid-based learning. To our knowledge, we are unaware of any works that offer an efficient way to discretize a projective space. On the contrary, the light-slab representation is compact and offers an easy discretization of the *Eucledian* ray space enabling grid learning.
>
>
> ***
> **Q3. Concerning the real-time mobile rendering claim**
>
> (a) We would like to draw attention to the fact that all our latency numbers are actually computed on mobile devices themselves (Tab. 3 main paper) leaving *no room for infeasibility on mobile devices*.
>
> (b) We agree with the reviewer that our `60-L W-256 LS` achieves 40 FPS on an iPhone 13, and this might be challenging to run on devices with lesser computational capabilities. We would kindly like to point out that this is a competitive FPS as compared to prior works. Further, this is exactly where our method comes into play with our `30-L W-256 LS` variant in rebuttal (which *also surpasses the MobileR2L visual fidelity*), delivering *~50 FPS* with *almost half FLOPs* than both `60-L W-256 LS` and `60-L W-256 MR2L`. Additionally, an even lighter variant, `15-L W-256 LS` in rebuttal with ~3.3x fewer FLOPs than full-sized models delivers a similar visual fidelity as that of the full-sized `60-L W-256 MR2L`.
>
> (c\) To numerically support the claim of our method's real-time performance on Android phones, we show the *on-device latency* on the **Snapdragon SM8450 chip** used in various **Android devices**, including **Huawei Honor Magic 4, OnePlus 10 Pro, Oppo Find X5 Pro, vivo iQOO 9, and Xiaomi 12**. We obtain competitive latency numbers (Tab. B) for our full-sized LightSpeed network and *much lower latency for our 30-layered network* which has better rendering fidelity than full-sized MobileR2L as shown in Tab. C of the original rebuttal.
>
>
> > Table B: **Rendering Latency Analysis on various scenes**: LightSpeed maintains a competitive rendering latency (ms) to MobileR2L on the Snapdragon SM8450 chip.
> >| Scenes  |  MobileR2L| Ours | Ours (30-L) |
> >| :------------ | :-----------: | :-----------: | :-----------: |
> >| LLFF  | 39.14 | 45.65 | 32.89|
> >| Synthetic $360^\circ$ | 40.86 | 41.26 | 33.87|
>
> (d) We are building a mobile application following MobileR2L for real-time demonstration that requires substantial software engineering efforts and time to build (please kindly notice that the code for the application from MobileR2L has not been released). We would be happy to share a real-time demonstration video; however, the application is not ready right now and given the *limited time remaining* for the discussion period, it is not feasible to build the on-device application on such a short notice.
>
> We are certain that the latency and FPS numbers we provide reflect the on-device performance of the method (we could provide CoreML benchmark reports for different chips in the anonymous link if the reviewer prefers). That being said, we would like to assure the reviewer that we will release a full-fledged demo once the mobile application is ready.
>
> ***
>
> We sincerely hope that our response answers your questions. It would be our great pleasure if you would consider updating your review or score.
>
>
> Best,
>
> Authors

---

> > ### Comment · Reviewer_bQpx · 2023-08-21
> >
> > I want to express my gratitude to the author for their thorough response to the concerns I highlighted.
> >
> > I am confident that the forthcoming revisions will address the areas of related work and representation selection. Additionally, I acknowledge the integration of grid representation and light slab representation as a meaningful contribution.
> >
> > **My chief reservations lie in the paper's somewhat audacious claims, particularly the assertions of "Light Speed" and operability "on mobile device.** The detailed performance metrics on both Qualcomm and Apple chips are indeed valuable, but I firmly believe that a video demonstration is indispensable to validate such bold statements. My assessment would be more favorable with further empirical evidence, perhaps at this juncture a benchmark on CoreML as proposed by the author. I'm aware of the time constraints and would like to clarify that my 'borderline' rating is not an outright call for rejection. Instead, it's a nudge for the author to ensure these highlighted areas receive the necessary emphasis in the revised manuscript.

---

> ### Author Response · Authors · 2023-08-21
>
> We thank the reviewer for their effort and the opportunity to further provide supporting evidence for our work. We share benchmarking images of LightSpeed CoreML packages done on Apple A15 and A16 chips as empirical proof of our method's real-time capabilities via an anonymous link shared to AC only (required by the NeurIPS PCs). We kindly ask the reviewer to get the link from AC.
>
> Best,
>
> Authors

---

### Author Rebuttal · Authors · 2023-08-09

We sincerely thank all the reviewers for their thoughtful comments and appreciate their findings that our novel method simplifies real-time novel view synthesis on mobile devices while performing better than exisiting works with high-quality and fast rendering even on mobile devices (bQpx, XtjZ, CSSm, BeXQ).


Specifically, our method is `noteworthy` in using a `traditionally underutilized light-slab representation` (bQpx) to formulate a grid-based ray-space for light field learning. This yields a`more compact and efficient ray representation` (bQpx) and replaces `the commonly used Plücker coordinates` (8eAH). We also `compress the ray space using 2D planes for efficienty` (8eAH). Our method further overcomes the limitations of light-slab representation by extending it to non-frontal $360^\circ$ scenes using a divide-and-conquer strategy (bQpx,BeXQ,8eAH). Our study provides `an extensive evaluation` (CSSm), demonstrating `clear results and advantages` (BeXQ) over prior methods. Further, our paper has `clear and easily understandable presentation` (CSSm).

We will incorporate all suggestions and sincerely hope this will help reviewers finalize their judgements.


***
We address common questions from reviewers in the following and answer other questions in the individual responses. We also upload the author response PDF to include more comparison figures.

**Q1. To bQpx, 8eAH. About comparison with SIGNET, NeuLF, and ProLiF.**

We thank the reviewers for pointing out these works. We propose a novel method to utilize grid-based representations for light field learning. To this end, we find the light-slab representation to be compact and easy to discretize for learning a grid-based ray space. To limit storage requirements, we decompose the 4D ray-space into six 2D planes. Further, we propose a divide-and-conquer strategy to utilize the light-slab representation, originally designed for frontal scenes, for modelling non-frontal scenes.

While all SIGNET, NeuLF, and ProLiFF use the light-slab representation, they differ significantly from LightSpeed.

- SIGNET[a] explores a different problem of compressing a given light field using ultra spherical network input encoding, with no guarantee for photo-realistic view synthesis, while our method targets and *achieves* real-time photorealistic view synthesis, specifically on mobile devices.
- NeuLF[b] fails to capture fine-level details of scenes due to the absence of any network input encoding and takes 70ms to render a 567x1008 frame on an RTX 3090. On the other hand, LightSpeed has no issues *learning fine-level detail*s, as shown in Fig. 4 in the main paper and *runs in real-time* on mobile devices.
- ProLiF[c] uses volumetric rendering to generate pixel values prohibiting real-time inference speeds. In contrast, LightSpeed runs on mobile-devices in *real-tim*e.

We will discuss these works in detail in the revised paper.

***
**Q2. To bQpx, CSSm. About results on unbounded scenes.**


The rendering fidelity of our approach (LightSpeed) is closely tied to the performance of the corresponding NeRF teacher. LightSpeed uses Instant NGP [d] teachers for both bounded and unbounded $360^\circ$ scenes to maintain experimental consistency. We would like to highlight that Instant NGP introduces the artifacts to unbounded scenes, which are carried forward to LightSpeed via the mined pseudo-data. We share some of the pseudo-data images from Instant-NGP in Fig. 3 of the rebuttal PDF. MipNeRF360 [e] specifically uses space contraction techniques to model the unbounded nature of the scene and deal with blurriness in the renderings. It further introduces a distortion-based regularizer to remove floater artifacts and prevent background collapse. The techniques introduced by MipNeRF360 tackle the same type of artifacts pointed out in the reviews. Hence, using MipNeRF360 teachers will mitigate both these issues and could boost the visual fidelity on unbounded scenes for LightSpeed.


***
**References:**

[a] Feng, Brandon Yushan, and Amitabh Varshney. "Signet: Efficient neural representation for light fields." ICCV. 2021.

[b] Li, Zhong, et al. "Neulf: Efficient novel view synthesis with neural 4d light field." arXiv. 2021.

[c] Wang, Peng, et al. "Progressively-connected Light Field Network for Efficient View Synthesis." arXiv. 2022.

[d] Müller, Thomas, et al. "Instant neural graphics primitives with a multiresolution hash encoding." ToG. 2022.

[e] Barron, Jonathan T., et al. "Mip-nerf 360: Unbounded anti-aliased neural radiance fields." CVPR. 2022.

---

### Decision · Program_Chairs · 2023-09-21

**Decision:**

Accept (poster)

**Comment:**

The manuscript received a range of evaluations from the reviewers, including one strong accept, two weak accepts, one borderline accept, and one borderline reject. It introduces an approach to parameterizing a neural light-field using the K-Planes concept. Initial concerns were raised regarding the ablation comparisons with Plücker coordinates and performance across various mobile platforms. The author addressed these concerns by presenting additional experiments in the rebuttal. The AC recommends acceptance of the manuscript. However, the AC agrees with the sentiment that a video demonstration showcasing real-time performance would significantly enhance the paper's impact. Therefore, the lack of such a demonstration hinders the recommendation for spotlight or oral presentation.